# Ruthenium(II) Complexes Coupled by Erianin via a Flexible Carbon Chain as a Potential Stabilizer of *c-myc* G-Quadruplex DNA

**DOI:** 10.3390/molecules28041529

**Published:** 2023-02-04

**Authors:** Zhixiang Wang, Wentao Liu, Guohu Li, Jiacheng Wang, Bin Zhao, Peishan Huang, Wenjie Mei

**Affiliations:** 1School of Pharmacy, Guangdong Pharmaceutical University, Guangzhou 510006, China; 2Department of Southern Pharmacy, Guangdong Jiangmen Chinese Medical, Jiangmen 510047, China; 3Guangdong Province Engineering Technology Centre for Molecular Probe and Bio-Medical Imaging, Guangzhou 510006, China

**Keywords:** ruthenium(II) complexes, erianin, *c-myc* G-quadruplex DNA, stabilizer

## Abstract

Herein, two novel ruthenium(II) complexes coupled by erianin *via* a flexible carbon chain, [Ru(phen)_2_(L_1_-(CH_2_)_4_-erianin)](ClO_4_)_2_ (L_1_ = 2-(2-(tri-fluoromethyphenyl))-imidazo [4,5*f*][1–10]phenanthroline (1) and [Ru(phen)_2_(L_2_-(CH_2_)_4_-eria)](ClO_4_)_2_ (L_2_ = 2-(4-(tri-fluoromethyphenyl))-imidazo [4,5*f*][1,10]phenanthroline (2), have been synthesized and investigated as a potential G-quadruplex(G4) DNA stabilizer. Both complexes, especially 2, can bind to *c-myc* G4 DNA with high affinity by electronic spectra, and the binding constant calculated for 1 and 2 is about 15.1 and 2.05 × 10^7^ M^−1^, respectively. This was further confirmed by the increase in fluorescence intensity for both complexes. Moreover, the positive band at 265 nm in the CD spectra of *c-myc* G4 DNA decreased treated with 2, indicating that 2 may bind to *c-myc* G4 DNA through extern groove binding mode. Furthermore, fluorescence resonance energy transfer (FRET) assay indicated that the melting point of *c-myc* G4 DNA treated with 1 and 2 increased 15.5 and 16.5 °C, respectively. Finally, molecular docking showed that 1 can bind to *c-myc* G4 DNA in the extern groove formed by base pairs G7–G9 and G22–A24, and 2 inserts into the small groove of *c-myc* G4 DNA formed by base pairs T19–A24. In summary, these ruthenium(II) complexes, especially 2, can be developed as potential *c-myc* G4 DNA stabilizers and will be exploited as potential anticancer agents in the future.

## 1. Introduction

G-quadruplex DNA, a secondary conformation of DNA molecules, plays a key role in DNA replication, transcription, and genomic maintenance [1]. Smart tactics have been developed to discover potential candidates through screening small molecules binding and stabilizing G4 DNA. In present research, there are three main therapeutic strategies to study G-quadruplexes as a promising target for cancer therapy. First, small molecular compounds bind to the G4 DNA at the telomere and stabilize its structure to affect the activity of telomerase. For example, several G4 ligands, such as telomatostatin [2], 2,6-diamineanthraquinone derivatives [3], and RHPS4 [4], can cause telomere dysfunction. Second, G4 ligand-specific binding with oncogenes such as *c-myc*, VEGF, and bcl-2 can affect its expression [5]. Third, G4 ligands binding with G4 DNA to enhance its gene instability may be used as a therapeutic method to induce tumor cell apoptosis and autophagy [6].

Recent research suggests that the G-rich sequence of the MYC oncogene can also form a G-quadruplex structure through a Hoogesteen hydrogen bond. *C-myc* G4 DNA is an important transcription factor overexpressed in 70% of human cancers. The NHE III1 (nucleic acid hypersensitivity element III1) in the promoter region of *c-myc* is rich in guanine and can form an intramolecular G-quadruplex conformation [7,8]. The aberrant expression of *c-myc* in cells causes several gene alterations, resulting in the occurrence and progression of cancers such as breast cancer, colon cancer, cervical cancer, and small cell lung cancer, among others [9]. *C-myc* promoter G-quadruplex has become one of the most concerned sequences in DNA due to its important role in cell growth, proliferation, apoptosis, senescence, and tumor formation.

A large number of compounds have been reported that are capable of interacting with and stabilizing *c-myc* G4 DNA, such as some G4-interacting ligands: 2,6-diamido anthraquinones, TmPyP410 [5], and PIPER [10]. After that, BRACO-19 [11], Se2SAP [12], BMSG-SH-3 [13], telomestatin, etc., have been designed and studied one after another. Hurley’s team reported quarfloxin (CX-3543), which selectively acts on the oncogene promoter *c-myc* G-quadruplex and is currently in phase II clinical study (ClinicalTrials.gov identifier: NCT00780663) [14]. Previous studies have shown that most ligands that bind to G4 DNA have a flat aromatic/heteraromatic core that interacts with the flat aromatic G-quadruplex in the G4 structure through hydrophobic and packing interactions. These studies have indicated that we should give more consideration to G4 DNA as a potential therapeutic target for cancer and it is necessary to design more small molecules that bind and interact with G4 DNA.

Ruthenium(II) complexes have been extensively studied as G4 DNA stabilizers [15,16,17,18]. For instance, a dinuclear ruthenium complex can bind firmly to the telomeric G-quadruplex and initiate and stabilize the development of an antiparallel G-quadruplex of telomeric DNA [19]. Moreover, a ruthenium complex Λ-[Ru(phen)_2_(qdppz)]^2+^ can bind with a modified human telomeric G-quadruplex sequence in an antiparallel chair topology [20]. Additionally, the bcl-2 G-quadruplex could be stabilized by ruthenium(II) complexes ([Ru(bpy)_2_(tip)]^2+^ and [Ru(phen)_2_(tip)]^2+^) [21]. Previously, our research group discovered that polypyridyl ruthenium(II) complexes can bind and stabilize the conformation of *c-myc* G4 DNA in groove binding mode, resulting in *c-myc* expression down-regulation [22]. The aryl alkyne group modified two ruthenium(II) complexes as potential luminescent switch-on probes for G4 DNA, with a stronger affinity for G4 DNA than double-stranded DNA [23]. Furthermore, two arene ruthenium(II) complexes [(*η*^6^-RC_6_H_5_)Ru(*m*-MOPIP)Cl]Cl (R = -H, 1; R = -CH_3_, 2) that act as small molecule inhibitors of *c-myc* G4 DNA have been identified, and both of these complexes demonstrate high affinity for *c-myc* G4 DNA in the groove binding mode [24]. In addition, we discovered that a series of phenanthroimidazole derivatives may stabilize the *c-myc* G4 DNA structure and that one of these phenanthroimidazole derivatives can cause CNE-1 cell death as nasopharyngeal cancer inhibitory agents [25]. As a possible inhibitor of *c-myc* G4 DNA, two polypyridyl ruthenium(II) complexes [Ru(bpy)_2_L](ClO_4_)_2_ (L = *p*-TEPIP (1) and *p*-BEPIP (2) containing alkynes can preferentially bind and stabilize *c-myc* G4 DNA [26]. Moreover, a chiral ruthenium(II) complex (∆-[Ru(bpy)_2_(DPPZ-R)](ClO_4_)_2_, R = -C≡C(C_6_H_4_)NH_2_) was reported as a potential *c-myc* G4 DNA stabilizer, inducing DNA damage to suppress triple-negative breast cancer progression [27]. Additionally, an arene ruthenium(II) complex, (*η*^6^-MeC_6_H_5_)Ru(o-ClPIP)Cl]Cl, exhibits moderated binding affinity to KRAS G4 DNA by groove mode [28]. What is more, a series of novel arene ruthenium (II) complexes with difluorinated ligands could bind to and stabilize *c-myc* G4 DNA [29].

Two novel ruthenium(II) complexes were successfully synthesized in this investigation using erainin, a bipyridyl molecule isolated from the traditional Chinese medicine dendrobiumchrysotoxum lindl, and coupled by a flexible carbon chain (Figure 1) [30]. It has been discovered that both complexes, particularly **2**, have a substantially higher binding affinity to *c-myc* G4 DNA and help to maintain the DNA’s shape. Moreover, molecular docking tests were carried out to clarify how **2** interacted with the *c-myc* G4 DNA. The further studies demonstrate that **2** can reduce *c-myc* expression *in vitro* by using the PCR-stop test. These findings suggest that **2** may act as a possible stabilizer of *c-myc* G4 DNA.

## 2. Results and Discussion

### 2.1. Synthesis and Characterization

The targeted complexes were prepared according to Figure 2. As usual, the target complexes **1**–**2** were obtained by heating the mixture of the intermediate **1b**–**2b** and erianin under the irradiation of microwave at 90 °C for 30 min. Complexes **1b**–**2b** was obtained by heating the mixture of **1a**–**2a,** which was obtained by reflux Ru(phen)_2_Cl_2_^.^2H_2_O with corresponding ligand (**L_1_** or **L_2_**) in the solvent of ethylene glycol, and 1,4-dibromobutane in DMF solution at 60 °C for 24 h. The chemical shifts of **1** and **2** in ^1^H NMR at 6.51 and 6.73 ppm can be attributed to the erianin benzene ring (Figure 1). The chemical shifts of **1** and **2** at 3.48, 3.57, and 3.70 ppm can be attributed to methoxy group on the erianin. For **1**, the chemical shifts at 1.77, 1.97, 3.82, and 4.61 ppm can be attributed to the flexible chain of the compound. The chemical shifts at 7.66, 8.03, and 9.03 ppm can be attributed to H_5_, H_6_, and H_7_ in the phenanthroimidazole ligand, respectively. The chemical shift attributed to the phenanthroline ring appeared at 8.79, 8.41, 8.09, and 7.79 ppm. For **2**, the chemical shifts at 7.94 and 8.13 ppm can be attributed to trifluoromethyl-benzene. In addition, the lipophilicity of these complexes was measured by an n-octanol/water partition coefficient (log*P_o/w_*) study (Appendix A). Two erianin-modified ruthenium(II) complexes **1** and **2** exhibited higher log*P_o/w_* values (−0.139 and −0.070, respectively) due to the lipophilic nature of their erianin group. In contrast, **1b** and **2b** observed lower log*P_o/w_* values (−0.869 and −0.71, respectively). These data imply that complexes **1** and **2** have higher lipophilicity than **1b** and **2b**. These results indicated that the introduction of erianin can improve the lipophilicity of ruthenium(II) complexes, making the lipophilicity of **1** and **2** higher than that of complexes **1b** and **2b**, which may contribute to the transmembrane uptake ability of ruthenium(II) complexes.

### 2.2. DNA Binding Behaviors

#### 2.2.1. Electronic Spectra Titration Experiments

Electron absorption spectroscopy was used to evaluate the binding behavior of ruthenium(II) complexes coupled with erianin, and their intermediates to *c-myc* G4 DNA were assessed. Usually, the characterized absorption in electronic spectra of ruthenium(II) undergoes significant hypochromism and red-shift effect, ascribed to the strong interaction between the complexes and DNA. The degree of the change depends on how strongly the binding to the DNA is accomplished.

As shown in Figure 2, the electronic spectra of **1** and **2** exhibit the typical MLCT (metal-to-ligand charge transfer) at about 454 and 453 nm as well as the typical IL (ligand internal charge transfer) absorption at about 261 and 263 nm, respectively. The results showed that the hypochromism in IL absorption was 12.3% and 25.4% for **1** and **2**, respectively, which was significantly larger than the hypochromism in IL absorption for the comparable intermediates **1b** and **2b**, which was 4.6% and 9.8%, respectively. Complex **1** exhibited a hypochromism of 9.4% at MLCT absorption, while **2** showed a hyperchromism of 16.6% at MLCT absorption. The hypochromism rate during MLCT absorption increased when compared to intermediates **1b** and **2b**, where the hypochromism rate during MLCT absorption was 3.5% and 7.9%, respectively. The results show that ruthenium(II) complexes treated with erianin have a greater planar area and thus a stronger binding capacity to *c-myc* G4 DNA than before the alteration.

#### 2.2.2. Fluorescence Response towards c-myc G-Quadruplex DNA

The fluorescence response of **1** and **2** to *c-myc* G4 DNA was studied by fluorescence spectra. As shown in Figure 3, ruthenium(II) complexes displayed a strong emission at 590 nm in buffer solution (10 mM Tris-HCl, 100 mM KCl, pH = 7.4) when excited at 466 nm. Moreover, a large Stokes shift (124 nm) was observed, which is beneficial for imaging analysis and more availably reduces the self-quenching and scattering light of the probe [31]. The fluorescence of the ruthenium(II) complexes (**1b** and **2b**) was not significantly changed by the addition of *c-myc* G4 DNA. In contrast, under the same experimental conditions for **1** and **2**, significant fluorescence changes were observed. Interestingly, when *c-myc* G4 DNA was titrated into solution of **2**, the fluorescence intensity increased gradually to about 1.5-fold of the original, suggesting that the compound could effectively intercalate into the hydrophobic region between the base pairs of *c-myc* G4 DNA, while the aromatic chromophore of the compound was protected from the quenching of solvent molecules. When *c-myc* G4 DNA was titrated into solution of **1**, the fluorescence intensity decreased gradually, indicating that the interaction between the compound and the solvent water causes the fluorescence intensity of the solution to decrease.

In addition, [Ru(bpy)_3_]^2+^ is used as the standard for the calculation of fluorescence quantum yield [32]. According to the formula, the results are shown in Appendix A. The results showed that the fluorescence quantum yield of **2** was the highest, which was consistent with the fluorescence titration experiment. Among the four complexes, we found that the fluorescence quantum yield of ruthenium(II) complexes was increased after modification with erianin, indicating that this modification had a certain improvement effect. The results show that **2** can can bind to *c-myc* G4 DNA with high affinity.

#### 2.2.3. Circular Dichroism (CD) Spectroscopy

Circular dichroism spectroscopy has been used to investigate the conformation change of G4 DNA in the presence of tiny disturbances. In general, there is almost no significant perturbation when small molecules bind to G4 DNA in noncovalent binding and even electrostatic groove binding mode, which can be ascribed to there being no influence on the secondary structure of G4 DNA molecules. When small molecules interact with G4 DNA, an intensity change and red- or blue-shift can be observed [33]. Figure 4 shows the CD spectra of *c-myc* G4 DNA in the absence and presence of **1** and **2**.

As seen in Figure 4, the CD spectra of the parallel G-quadruplex conformation-forming G4 DNA structure made from oligonucleotides pu22 had a positive peak at about 264 nm and a negative peak at approximately 243 nm [30,31]. No obvious spectral changes were seen in the CD spectrum with the addition of **1b** and **1** ([*c-myc*]/[Ru] = 1:6), showing that the interaction between complexes and *c-myc* G4 DNA barely altered the G-quadruplex structure in a K^+^ solution. The spectra after the addition of **2b** and **2** demonstrate that the binding of **2b** and **2** resulted in a minor reduction in or enhancement of the positive band in a K^+^ solution at 265 nm. Additionally, a positive induced CD signal was seen in the 290–300 nm region as the concentration of complexes **2b** and **2** increased. These findings suggested that **2b** and **2** may connect to the *c-myc* G4 DNA via electrostatic interaction or straightforward groove binding.

#### 2.2.4. Melting FRET and Competitive FRET Assays

The melting point of *c-myc* G4 DNA in the presence of **1** and **2** was examined using the FRET (Fluorescence resonance energy transfer) experiment, demonstrating the importance of maintaining the conformation of the G-quadruplex. In fact, the thermal behaviors of DNA in the presence of complexes can provide details about the structural changes that occur when the temperature is elevated, as well as specifics about the potency of interactions between complexes and DNA. An intercalative method of metal complex binding to DNA is indicated by a high Tm value, whereas an electrostatic or groove method is indicated by a low value (1–3 °C) [34].

As shown in Figure 5, the melting point of *c-myc* G4 DNA increased after the addition of **1**, **1b**, **2**, and **2b**, and the ΔTm values for **1**, **1b**, **2**, and **2b** are about 8.0, 13.1, 15.5, and 16.5 °C, respectively. These results suggested that the large DNA-binding affinity of the complexes, especially **2**, displayed better stability than other complexes, which accords with the aforementioned results. Moreover, the results also show that **1** and **2** exhibit better stability than **1b** and **2b**, which means ruthenium(II) complexes can stabilize the G4 DNA better after being modified by erianin via a flexible carbon chain. Moreover, the competitive FRET assay was also performed to confirm the selectivity of arene ruthenium(II) complexes between *c-myc* G4 DNA and ds26 duplex DNA. In the system of *c-myc* and complex, ds26 was added as the competitor. As shown in Figure 5, *c-myc* G4 DNA was affected to varying degrees in the case of excessive ds26 (10-fold, 20-fold, and 40-fold), among which **1** and **2** selectively stabilized *c-myc* G4 DNA, and, as shown in Figure 5E, the selectivity of **1** and **2** for G4 DNA is better than that of **1b** and **2b**, indicating that the ruthenium(II) complexes coupled by erianin improve the selectivity of G4 DNA. These results indicated that erianin-modified ruthenium(II) complexes can selectively bind and stabilize *c-myc* G4 DNA and the introduction of erianin improves the selective stability of the ruthenium(II) complexes to *c-myc* G4 DNA.

#### 2.2.5. PCR-Stop Assay

PCR-stop assays, using templates containing G-quadruplex sequences, have been used by several researchers to demonstrate ligand binding to G-quadruplexes [35]. We investigated the effects of ruthenium(II) complexes on *c-myc* G-quadruplex stabilization by the PCR-stop assay. Using Taq polymerase as catalyst, the *c-myc* template and corresponding primer *c-myc rev* were combined to form double-stranded DNA. When ruthenium(II) complexes are present in the reaction system, they will promote or stabilize *c-myc* G4 DNA and block its hybridization, resulting in the halt of PCR reaction.

As shown in Figure 6, the polymerization extension of *c-myc* and *c-myc rev* was inhibited in the presence of **1** and **2**. In the presence of the same concentration of **1** and **2**, the polymerization elongation of *c-myc* and *c-myc rev* is almost not inhibited. These results indicate that the erianin-modified ruthenium(II) complexes **1** and **2** are able to induce and stabilize *c-myc* G-quadruplex structures and inhibit the amplification of PCR products.

#### 2.2.6. Molecular Docking

To gain further insight into the interaction of G4 DNA with ruthenium(II) complexes, molecular docking studies were carried out. For this purpose, molecular docking studies of **1** and **2** with *c-myc* G4 DNA of sequence 5′-TGAGGGTGGGTAGGGTGGGTAA-3′ (PDB ID:2L7V) were performed to validate the chosen binding mode and binding site [36].

As shown in Figure 7, the complexes **1** and **2** bind to *c-myc* G4 DNA mainly by electrostatic interaction mode, and the binding energy is about −6.24 and −5.62 kcal/mol, respectively. In addition, complex **1** can insert into the groove of *c-myc* G4 DNA, formed by base pairs G7–G9 and G22–A24. The auxiliary ligand of complex **2** can insert into the small groove of *c-myc* G4 DNA, formed by base pairs T19–A24. Thus, we speculated that there are electrostatic interactions and groove binding modes between ruthenium(II) complexes and *c-myc* G4 DNA.

## 3. Materials and Methods

### 3.1. Chemicals

All reagents were purchased from commercial suppliers and used without further purification. All buffers were prepared using double-distilled water and used in all experiments. *C-myc* Pu22 (TGAGGGTGGGTAGGGTGGGTAA) was purchased from General biosystems (Anhui) Co., Ltd (Chuzhou, China). Pu22 was prepared with a concentration of 100 μM using 10 mM Tris-buffer (pH = 7.2, 100 mM KCl). Before the experiment, it was heated at 95 °C for 5 min to denaturate, and then gradually cooled to room temperature and placed in a refrigerator at 4 °C overnight to form G-quadruplex. 1,10-phenylline and 2,2′-bipyridine from Sigma-Aldrich (St. Louis, MO, USA). 1,4-dibromobutane from Aladdin reagent (Shanghai, China) Co., Ltd. Ligand and ruthenium(II) complexes were prepared adopting the reported procedure.

### 3.2. Instruments

By using an Anton Paar Monowave 300 microwave reactor, the complexes were synthesized. On a Shimadzu UV2550 spectrophotometer, electronic absorption spectra were captured. Fluorescence studies were performed on an RF-5301PC luminescence spectrophotometer (Shimadzu, Japan). NMR (nuclear magnetic resonance) spectra (500 MHz) were recorded on a Bruker instrument. Electrospray ionization mass spectra (ESI-MS) were recorded on an Agilent 1100 LC-MS. FRET melting and competitive assays were recorded with a Bio-Rad real time PCR (CFX96 Touch). Mitsuwa Chemicals supplied the ruthenium(III) chloride hydrate (Tokyo, Japan).

### 3.3. Synthesis and Characterization of [Ru(phen)_2_(L_1_(CH_2_)_4_Br)] (ClO_4_)_2_ (1b)

Synthesis of [Ru(phen)_2_(L_1_(CH_2_)_4_Br)] (ClO_4_)_2_ (**1b**) was synthesized following the literature, but with some modifications. A mixture of **1a** (207 mg, 0.2 mmol), dried K_2_CO_3_ (2 g), and DMF (15 mL) was activated for 30 min at 60 °C. 1,4-dibromobutane (1 mL) was added and the reaction continued at 60 °C for 24 h. After the reaction liquid was cooled, the filter liquor was diluted with water, and methyl tert-butyl ether was extracted three times to collect the aqueous phase. Adding sodium perchlorate into the aqueous phase produces a large amount of orange suspended solid and collects orange suspended solid, dried under vacuum, and purified by Al_2_O_3_ column chromatography with acetonitrile/methylbenzene (1:1 *v/v*) as eluent, the yield was 53%. ESI-MS (in CH_3_CN, *m/z*): 481.0536 ([M-2ClO_4_^−^]^2+^, Cal: 480.40); 539.0770 ([M-ClO_4_^−^ + NH_4_^+^]^2+^, Cal: 539.055). ^1^H NMR (600 MHz, DMSO) δ 9.02 (ddd, J = 24.1, 15.9, 4.9 Hz, 2H), 8.80 (tdd, J = 5.4, 4.7, 2.6 Hz, 4H), 8.41 (d, J = 3.1 Hz, 4H), 8.08 (ddd, J = 7.3, 5.7, 2.6 Hz, 6H), 8.00–7.92 (m, 2H), 7.86–7.74 (m, 8H), 4.55 (s, 2H), 3.43 (dd, J = 10.2, 3.7 Hz, 2H), 1.96–1.81 (m, 4H). ^13^C NMR (151 MHz, DMSO) δ 153.45 (s), 153.11 (s), 152.05 (s), 151.54 (s), 150.83 (s), 147.62 (s), 146.37 (s), 146.19 (s), 137.33 (s), 136.87 (s), 133.35 (s), 133.17 (s), 131.99 (s), 130.89 (s), 130.54 (s), 129.51 (s), 129.30 (s), 128.56 (s), 127.89 (s), 127.49 (s), 127.21 (s), 126.88 (s), 126.66 (s), 126.02 (s), 125.73 (s), 125.03 (s), 123.22 (s), 121.85 (s), 65.24 (s), 51.46 (s), 34.67 (s), 29.20 (s), 28.22 (s), 21.85 (s).

### 3.4. Synthesis and Characterization of [Ru(phen)_2_(L_2_(CH_2_)_4_Br)] (ClO_4_)_2_ (2b)

Synthesis of [Ru(phen)_2_(L_2_(CH_2_)_4_Br)] (ClO_4_)_2_ (**2b**) was obtained using the same method as above, but used **2a** (207 mg) instead, yield: 49%. ESI-MS (in CH_3_CN, *m/z*): 481.0523 ([M-2ClO_4_^−^]^2+^, Cal: 480.40); 539.0752 ([M-ClO_4_^−^ + NH_4_^+^]^2+^, Cal: 539.055); 597.0988 ([M + 2NH_4_^+^]^2+^, Cal: 597.055). ^1^H NMR (500 MHz, DMSO) δ 9.11–9.03 (m, 2H), 8.83–8.77 (m, 4H), 8.41 (s, 4H), 8.17–8.02 (m, 10H), 7.88–7.73 (m, 6H), 4.82 (d, J = 6.6 Hz, 2H), 3.57–3.39 (m, 2H), 2.04–1.91 (m, 2H), 1.88–1.71 (m, 2H). ^13^C NMR (126 MHz, DMSO) δ 152.72 (s), 151.86 (s), 151.18 (s), 151.01 (s), 149.49 (s), 148.68 (s), 145.58 (s), 145.45 (s), 144.20 (s), 135.19 (s), 129.16 (s), 128.79 (s), 128.51 (s), 127.25 (s), 126.50 (s), 125.09 (s), 124.66 (s), 124.54 (s), 124.32 (s), 123.89 (s), 121.26 (s), 119.99 (s), 107.89 (s), 65.10 (s), 32.86 (s), 27.03 (s), 22.98 (s).

### 3.5. Synthesis and Characterization of [Ru(phen)_2_(L_1_(CH_2_)_4_eria)] (ClO_4_)_2_ (1)

Synthesis of [Ru(phen)_2_(L_1_(CH_2_)_4_eria)] (ClO_4_)_2_ (**1**) was synthesized following the literature, but with some modifications. Erianin (31.8 mg, 0.1 mmol), **1b** (107 mg, 0.1 mmol), sodium methoxide (54 mg, 1 mmol), and DMF (15 mL) were added to a 30 mL quartz reaction tube. N_2_ was added at room temperature for 10 min, then irradiated by microwaves for 30 min at 90 °C. Following the reaction, 100 mL of water was added to dilute and sodium perchlorate was added. The mixture was then allowed to stand, and the resulting crude product filtered. The raw material was purified using a neutral alumina column, then the red band was collected by eluting with toluene and acetonitrile, and the solvent was removed by vacuum distillation to yield a red solid chemical, yield: 38%. ESI-MS (in CH_3_CN, *m/z*): 599.1642 ([M-2ClO_4_^−^]^2+^, Cal: 599.1650); 657.1872 ([M-ClO_4_^−^ + NH_4_^+^]^2+^, Cal: 657.665); 399.7785 ([M-2ClO_4_^−^ + H^+^]^3+^, Cal: 399.7766). For C_66_H_55_F_3_N_8_O_5_Ru: calcd: C, 55.31; H, 4.15; N, 7.82; found: C, 55.04; H, 4.17; N, 7.89. ^1^H NMR (500 MHz, DMSO) δ 9.06–9.00 (m, 2H), 8.82–8.77 (m, 4H), 8.42 (t, J = 5.6 Hz, 4H), 8.13–8.01 (m, 6H), 7.91–7.75 (m, 9H), 7.67 (dd, J = 8.5, 5.3 Hz, 1H), 6.88–6.62 (m, 3H), 6.50 (d, J = 11.9 Hz, 2H), 4.89–4.30 (m, 2H), 3.87–3.77 (m, 2H), 3.71 (d, J = 11.4 Hz, 6H), 3.59 (d, J = 13.6 Hz, 3H), 3.54–3.38 (m, 3H), 2.77 (d, J = 14.7 Hz, 4H), 2.05–1.88 (m, 2H), 1.74 (t, J = 21.7 Hz, 2H). ^13^C NMR (126 MHz, DMSO) δ 153.45 (s), 153.09 (s), 152.02 (s), 151.53 (s), 150.77 (s), 148.00 (s), 147.60 (s), 146.37 (s), 146.17 (s), 137.78 (s), 137.35 (s), 136.87 (s), 136.03 (s), 134.54 (s), 133.15 (s), 133.01 (s), 131.83 (s), 130.94 (s), 130.47 (s), 128.54 (s), 127.98 (s), 127.41 (s), 127.22 (s), 126.89 (s), 126.46 (s), 126.05 (s), 125.71 (s), 125.24 (s), 123.01 (s), 121.86 (s), 120.93 (s), 114.09 (s), 112.51 (s), 110.01 (s), 106.11 (s), 67.98 (s), 60.40 (s), 56.04 (s), 46.63 (s), 38.14 (s), 37.06 (s), 27.09 (s), 25.90 (s).

### 3.6. Synthesis and Characterization of [Ru(phen)_2_(L_2_(CH_2_)_4_eria)] (ClO_4_)_2_ (2)

Synthesis of [Ru(phen)_2_(L_2_(CH_2_)_4_eria)] (ClO_4_)_2_ (**2**) was obtained in the same method as above, but used **2b** (107 mg) instead, yield: 42%. ESI-MS (in CH_3_CN, *m/z*): 599.1655 ([M-2ClO_4_^−^]^2+^, Cal: 599.1650); 657.1889 ([M-ClO_4_^−^ + NH_4_^+^]^2+^, Cal: 657.665); 715.2131 ([M + 2NH_4_^+^]^2+^, Cal: 716.165). For C_66_H_55_F_3_N_8_O_5_Ru: calcd: C, 54.61; H, 4.03; N, 7.71; found: C, 54.33; H, 4.08; N, 7.88. ^1^H NMR (500 MHz, DMSO) δ 9.10–9.03 (m, 2H), 8.81–8.77 (m, 4H), 8.42–8.39 (m, 4H), 8.13–8.05 (m, 9H), 7.94 (d, J = 8.3 Hz, 1H), 7.84–7.76 (m, 6H), 6.86–6.68 (m, 3H), 6.50 (d, J = 15.0 Hz, 2H), 4.90–4.78 (m, 2H), 3.85 (d, J = 6.6 Hz, 2H), 3.74–3.69 (m, 6H), 3.66 (s, 1H), 3.58 (d, J = 12.6 Hz, 3H), 3.48 (s, 2H), 2.76 (d, J = 15.9 Hz, 4H), 1.74 (dd, J = 13.3, 6.4 Hz, 2H), 1.63 (dd, J = 21.8, 6.2 Hz, 2H). ^13^C NMR (126 MHz, DMSO) δ 154.91 (s), 153.91 (s), 153.31 (s), 153.08 (s), 151.61 (s), 150.78 (s), 148.16 (s), 148.02 (s), 147.69 (s), 147.57 (s), 146.41 (s), 146.26 (s), 137.78 (s), 137.32 (s), 136.03 (s), 134.61 (s), 133.97 (s), 131.47 (s), 131.28 (s), 131.18 (s), 130.96 (s), 130.57 (s), 128.56 (s), 127.21 (s), 126.84 (s), 126.73 (s), 126.62 (s), 126.37 (s), 126.04 (s), 125.53 (s), 123.35 (s), 122.10 (s), 120.99 (s), 120.89 (s), 118.84 (s), 114.30 (s), 114.18 (s), 112.56 (s), 106.09 (s), 67.70 (s), 60.40 (s), 56.08 (s), 46.85 (s), 38.15 (s), 37.09 (s), 27.12 (s), 25.46 (s).

### 3.7. Electronic Spectra Titration Experiments

Electron absorption spectroscopy was performed at room temperature to detect the binding affinity between *c-myc* G4 DNA and the ruthenium(II) complex. The UV absorption curve of the ruthenium(II) complex (20 μM) in 10 mM Tris-HCl 100 mM KCl buffer solution was determined. Electron absorption titrations were recorded by varying the concentration of *c-myc* G4 DNA and in the 200–800 nm range.

### 3.8. Fluorescence Spectra

The fluorescence spectra of the ruthenium(II) complexes were titrated by successive additions of a stock solution of DNA. A solution of 3 mL 10 mM ruthenium(II) complex was added into the fluorescence colorimetric dish, and the fluorescence spectra of the complex at 500–750 nm were excited with the MLCT peak as the excitation wavelength. A 2 μL measure of *c-myc* G4 DNA was added to the cube each time, and the scanning was performed after 2 min of mixing to observe the change in the fluorescence emission peak of the ruthenium(II) complex with the addition of DNA. When the fluorescence intensity of the optimal fluorescence emission peak of the ruthenium(II) complex did not change significantly, the drip-addition of DNA was terminated.

### 3.9. Fluorescence Quantum Yields

Fluorescence quantum yield (Φ) refers to the ratio of the number of fluorescence photons emitted by a fluorescent substance after absorbing light to the number of photons absorbed by the excited light, and is an important parameter representing the fluorescence properties of a substance. Fluorescence intensity, and absorbance of the solution of the substance and the control substance, were measured at the same excitation wavelength. The calculation formula is as follows:$$\Phi_{\mathrm{unk}}=\Phi_{\mathrm{std}}\left(\frac{I_{\mathrm{unk}}}{A_{\mathrm{unk}}}\right)\left(\frac{A_{\mathrm{std}}}{I_{\mathrm{std}}}\right){\left(\frac{\eta_{\mathrm{unk}}}{\eta_{\mathrm{std}}}\right)}^{2}$$

The fluorescence quantum yields of the sample and standard are represented in the formula by the letters unk and std, respectively. Iunk and Istd stand for the integral regions of the sample’s and standard’s fluorescence spectra, respectively. Aunk and Astd stand for the sample’s and the standard’s respective ultraviolet absorbances at the sample’s optimal excitation wavelengths, whereas unk and std are the sample’s and the standard’s respective indexes of refraction. The standard in the current study was [Ru(bpy)_3_]^2+^ (in CH_3_CN, Φ = 0.062).

### 3.10. Circular Dichroism (CD) Spectroscopy

All circular dichroic chromatographic curves were collected on the Jasco I810 circular dichroic chromatograph. The ruthenium(II) complex (0–10 μM) was gradually added to 2 μM) *c-myc* G4 DNA at 25 °C, and the mixture was blown and balanced for 5 min until the curve changes could not be detected by light. The spectral curves in the range of 200–600 nm were collected at the scanning rate of 200 nm/min. The baseline was subtracted from the sample spectral curve by averaging two scans of the spectrum. Data analysis was performed using Origin9.0 software.

### 3.11. FRET Melting and Competitive FRET Assays

After being diluted in 10 mM Tris-HCl (pH 7.4), 100 mM KCl, and 10 mM Na3AsO4, fluorescently tagged oligonucleotide *c-myc* G4 DNA (5′-FAM-TGAGGGTGGGTAGGGTGGGTAA-TAMRA-3′) was annealed by being heated to 90 °C for 5 min, then slowly cooled to room temperature, 4 °C overnight. With a total reaction volume of 25 μL, 0.2 μM of labeled oligonucleotide *c-myc* G4 DNA, and various concentrations of complexes, the fluorescence curves of FAM at 30–100 °C were observed using a Bio-Rad real-time PCR (CFX96 Touch) detection system. The quantities of *c-myc* G4 DNA and the ruthenium(II) complexes were also held constant in the competitive FRET tests, and ds26 duplex DNA was used as a competitive binder to test the ruthenium(II) complexes’ capacity to bind to *c-myc* G4 DNA in a selective manner. Using Origin9.0 (Origin Lab Corp. Northampton, America), the final data analysis was carried out.

### 3.12. Polymerase Chain Reaction (PCR-Stop) Assay

The PCR-stop experiment was carried out in a 25 μL volume of 10 × PCR buffer containing *c-myc* (4 μM), *c-myc rev* (4 μM), Taq DNA polymerase (2.5 U), dNTPs (160 μM), and various complex concentrations (0, 10, 20, 40, and 80 μM). The above reaction solution was incubated in the PCR machine for 3 min at 95 °C, followed by 30 repeated cycles at 94 °C for 30 s, 58 °C for 30 s, and 72 °C for 30 s. Loading the amplified products on 15% native polyacrylamide gels in 1 × TBE buffer and run for 80 min at 120 V. The polyacrylamide gel was silver-stained and photographed by TILON 600 Imager System.

### 3.13. Molecular Docking

The molecular structure of two ruthenium(II) complexes were optimized using the ADF2019.104 suite program with the GGA: BP86 level of theory and the Mopac method, and the initial PDB structures were produced using Mercury software. We employed the Lamarckian genetic algorithm local search strategy with Auto-Dock 4.2.A to identify the binding mode and binding location of ruthenium(II) complexes in c-myc G4 DNA. The crystal structure of G4 DNA was downloaded from the Protein Data Bank (PDB ID:2L7V) by removing the other subunits of the structure keeping only the A-strand. The Gasteiger charge and other settings are assigned using the AutoDock tool. The DNA active site (x = 2.579, y = −0.627, and z = −4.749) was in the center of the grid box, which was made up of 126 × 126 × 126 points separated by 0.375 Å. Budgeting the binding affinities of each ligand atom was performed using Autogrid, and molecular docking simulation was performed using AutoDock 4.2. The conformation with the lowest binding free energy and the greatest number of cluster members was chosen as the most likely binding conformation. The docking results were visualized using Pymol.

## 4. Conclusions

In conclusion, the binding ability and stability of the ruthenium(II) complex coordinates to *c-myc* G4 DNA was enhanced by the introduction of the natural molecule erianin-modified. It has been discovered that both complexes, particularly **2**, have a relative higher affinity for the *c-myc* G4 DNA. Additionally, it has been demonstrated that both complexes **1** and **2** may externally attach to and stabilize *c-myc* G4 DNA, as well as down-regulate *c-myc* production in vitro. In other words, these ruthenium(II) complexes, notably **2**, have the potential to be employed in the future as *c-myc* G4 DNA stabilizers and anticancer agents.

## Data Availability

Data for the compounds are available from the authors.

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
