# Peer review of "Ruthenium(II) Complexes Coupled by Erianin via a Flexible Carbon Chain as a Potential Stabilizer of c-myc G-Quadruplex DNA"

_molecules, 2023, doi:10.3390/molecules28041529_

Round 1

Reviewer 1 Report

The authors describe the “Ruthenium(II) complexes coupled by erianin via flexible carbon chain as potential stabilizer of c-myc G-quadruplex DNA”.  Interestingly, molecular docking showed that 1 can bind to c-myc G4 DNA in the extern groove formed by base pairs G5-G11, and 2 insert into the small groove of c-myc G4 DNA formed by base pairs G18-G21. In summary, these ruthenium(II) complexes, especially 2 can be developed as potential c-myc G4 DNA stabilizer. Manuscript was well organized, and data provided is consistent. this would be reasonable to accept for publication in Molecules with minor revision.

I suggest authors to provide integration for both Ruthenium complexes in Figure 1.

If possible, Provide the dotted line in between two spectra in Figure 1 (Just for good separation).

Reviewer 2 Report

Dear authors,

It was with great interest that I was reading your manuscript. The manuscript is well written and the methods very well described. Additionally the references are relevant and the discussion supports the results. I only have two-three minor comments.

1. Despite the fact that the introduction is well structured, you need to improve it with newest examples of ruthenium complexes from the literature. 2021, 2022 et.c.

2. Moreover conclusions need to be written again. The complexation of metal ions with natural product derivatives is not a novel procedure. So please avoid writting something like this.

3. Proofreading is also needed.

4. Are the docking results evaluated by the essays?

5. Was it possible to gain a crystal structure from at least one of the complexes?

Best regards,
